# Peripheral Alterations Affect the Loss in Force after a Treadmill Downhill Run

**DOI:** 10.3390/ijerph18158135

**Published:** 2021-07-31

**Authors:** Nicola Giovanelli, Mirco Floreani, Filippo Vaccari, Stefano Lazzer

**Affiliations:** School of Sport Science, Department of Medicine, University of Udine, 33100 Udine, Italy; mirco.floreani@uniud.it (M.F.); filippo.vaccari@live.com (F.V.); stefano.lazzer@uniud.it (S.L.)

**Keywords:** trail running, muscle fatigue, voluntary activation, neuromuscular fatigue

## Abstract

Downhill running has an important effect on performance in trail running competitions, but it is less studied than uphill running. The purpose of this study was to investigate the cardiorespiratory response during 15 min of downhill running (DR) and to evaluate the neuromuscular consequences in a group of trail runners. Before and after a 15-min DR trial (slope: −25%) at ~60% of maximal oxygen consumption (V̇O_2_max), we evaluated maximal voluntary contraction torque (MVCt) and muscle contractility in a group of seventeen trail running athletes. Additionally, during the DR trial, we measured V̇O_2_ and heart rate (HR). V̇O_2_ and HR increased as a function of time, reaching +19.8 ± 15.9% (*p* < 0.001; ES: 0.49, *medium)* and +15.3 ± 9.9% (*p* < 0.001; ES: 0.55, *large*), respectively, in the last minute of DR. Post-exercise, the MVCt decreased (−22.2 ± 12.0%; *p* < 0.001; ES = 0.55, large) with respect to the pre-exercise value. All the parameters related to muscle contractility were impaired after DR: the torque evoked by a potentiated high frequency doublet decreased (−28.5 ± 12.7%; *p* < 0.001; ES: 0.61, *large*), as did the torque response from the single-pulse stimulation (St, −41.6 ± 13.6%; *p* < 0.001; ES: 0.70, *large*) and the M-wave (−11.8 ± 12.1%; *p* < 0.001; ES: 0.22, *small*). We found that after 15 min of DR, athletes had a decreased MVCt, which was ascribed mainly to peripheral rather than central alterations. Additionally, during low-intensity DR exercise, muscle fatigue and exercise-induced muscle damage may contribute to the development of O_2_ and HR drift.

## 1. Introduction

Downhill running (DR) has an important effect on performance in trail running competitions. In contrast to uphill locomotion, during DR, the metabolic demand is lower when utilizing the same (but inverted) mechanical power, and the efficiency is much higher [1,2]. Moreover, DR performance is more strenuous on the musculoskeletal structures [3], and it is largely affected by neuromuscular characteristics [4,5].

Some authors [1] compared the effects of uphill running vs. DR at ±10% slopes at ~60% of maximal oxygen uptake (V̇O_2_max) and reported that DR leads to considerably greater muscle damage, as shown by elevated levels of creatine kinase activity and interleukin 6 levels. Whatever the intensity, repeated eccentric exercise leads to an increase in metabolic demand (e.g., drift in V̇O_2_) [6,7,8]. Some authors [2,6] ascribed the V̇O_2_ drift to an increase in motor unit recruitment caused by muscle/connective tissue damage and local muscle fatigue. V̇O_2_ drift may be attributed to the higher neural input needed to maintain the required force [9], which, in turn, may provoke higher V̇O_2_ [10].

One of the effects of repeated eccentric exercise (e.g., downhill running) is an acute loss of strength that might persist for many days (i.e., even 7 to 10 days) after the end of the effort [2,11]. The ability to maximally generate muscle force and power might become impaired during and after exercise, which is ascribed to neuromuscular fatigue [12,13].

The deterioration in performance might be caused by alterations originating from any of the structures above the neuromuscular junction (i.e., central fatigue) and/or more locally from impairments occurring within the muscle itself (i.e., peripheral fatigue). Some authors [14] reported that central factors negatively limit the expression of force after a mountain ultra-marathon, in which DR sections are common. Furthermore, Giandolini et al. [4,5] observed that both central and peripheral alterations negatively affect the knee extensor (KE) maximal voluntary contraction torque (MVCt) after a downhill trail run. This last study was performed on a field and provided new information about DR in real-life conditions. However, during this study, the authors did not collect cardiorespiratory parameters. Hence, more data might be required to investigate how cardiorespiratory and neuromuscular systems together can cope with DR performance.

Thus, the aim of this study was to measure the cardiorespiratory parameters during 15 min of downhill running at a −25% slope at ~60% of V̇O_2_max and to evaluate the neuromuscular consequences using a previously proposed methodology [15]. Based on previous studies [4,15], we hypothesized that after DR, the MVCt expressed by the lower limbs would be lower because of peripheral alterations. Additionally, we expected to find a positive relationship between the loss in MVCt and the V̇O_2_ drift.

## 2. Materials and Methods

### 2.1. Participants

Seventeen healthy male trail runners (age: 33.9 ± 6.9 y, weight: 71.7 ± 7.8 kg, height: 1.76 ± 0.07 m, V̇O_2_max: 61.3 ± 6.4 mL/kg/min, ITRA performance index: 620 ± 124) participated in this study. The protocol was approved by the Ethics Committee of the University of Udine (CEUR-2017-Sper-033-Uniud), and it was conducted in accordance with the Declaration of Helsinki. Before the study began, we explained the purpose and objectives to each participant, and they provided informed consent.

### 2.2. Design and Procedures

Participants visited the laboratory two times (Figure 1). During the first day, we collected anthropometric data (stature and body mass), and the athletes became familiarized with all the procedures. Then, they performed an uphill incremental running test on a treadmill for measuring the maximal oxygen uptake (V̇O_2_max) and the maximal vertical velocity (V_vert_max). Seven days after, their second visit was dedicated to the measurements of neuromuscular characteristics before and after a 15-min downhill running trial at a −25% slope at ~60% of V̇O_2_max. In particular, before and 10 min after DR, we evaluated MVCt, voluntary activation (VA) and muscle contractility [16]. We acknowledge that 15-min of DR might be a short period of time, especially for trail runners. However, we performed some pilot tests and we saw that it was the best compromise for all level of participants that we enrolled.

Incremental uphill running test. V̇O_2_max, maximal heart rate (HR_max_) and V_vert_max were determined during a graded exercise test on a treadmill (Saturn, HP Cosmos, Nußdorf, Germany) under medical supervision. We decided to adopt an uphill test because we wanted to detect V̇O_2_max and thresholds, and it was more specific for this population, even if the subsequent test would be performed downhill. Additionally, it would not make sense to purpose a downhill running test since it has been reported that during DR, athletes do not reach V̇O_2_max [17]. After a 10-min warm-up at a self-selected speed, the athletes started the test at a speed of 6 km/h and a slope of 10%. Every two minutes, the slope increased by 2% until a maximum of 24%, which is the maximum slope that the treadmill can reach. After this step, the speed increased by 0.5 km/h until the subject experienced volitional exhaustion. We choose this protocol because it allowed us to increase the vertical velocity linearly by ~192 m/min every two minutes. During the test, we measured V̇O_2_ and carbon dioxide production (V̇CO_2_) breath-by-breath using a metabolic unit (Quark CPET, Cosmed, Italy). We calibrated the volume and gas analyzers before every test using a 3-L calibration syringe and calibration gas (16.00% O_2_ and 4.00% CO_2_), respectively. During the tests, we recorded HR with a dedicated device (Garmin, Lenesa, KS, USA).

Maximal voluntary contraction. Participants were seated on a customized seat [18], and the upper body was fixed by a belt (OMP racing, Italy) to prevent movement. An operator adjusted the seat to maintain the hip and knee angles at 90°. We attached the dominant leg (i.e., right leg) to a force sensor (AM C3, Laumas, Montechiarugolo, Italy) with a Velcro strap, and we acquired the data by using a dedicated device (MP 100, Biopac Systems Inc., Goleta, CA, USA) and software (Acqknowledge, Biopac Systems Inc., Goleta, CA, USA). We asked the participants to perform a set of warm-up submaximal contractions (8–10 trials) of increasing intensity with the knee extensor muscles of the right leg. Then, they performed a series (max 3 to 4 attempts) of maximal voluntary isometric contractions (MVCs) with a recovery time of three minutes between each trial. The MVC was defined as the highest value with less than a 5% difference from the second highest value of the series. Then, we calculated the MVCt (Nm) as the product of the maximal force (N) and the lever arm (m) at which that force was generated. During MVC, we collected the electromyographic (EMG) activities of a representative knee extensor (i.e., the vastus lateralis, VL).

### 2.3. Voluntary Activation and Muscle Contractility

We assessed voluntary activation (VA) by adapting the twitch interpolation technique proposed by Merton [19]. The procedure consisted of electrical stimulation administered during MVC to active the muscles and immediately after the end of the MVC when the muscles were relaxed. Electrical stimulation was provided superficially to the femoral nerve via a cathode electrode (circular shape, diameter: 1 cm) placed at the level of the inguinal triangle. The anode electrode (53 × 96 mm rectangular shape, PG472W Fiab, Firenze, Italia) was positioned over the gluteal fold. Electrical square-wave stimuli (pulse duration: 1 ms) were delivered by a constant current stimulator (DS7A, Digitimer, Hertfordshire, UK). Stimulation intensity was set to be approximately 130% of the maximal recruitment intensity and ranged from 40 mA to 80 mA. A square-wave pulse of 1-ms duration was administered at a maximal stimulator voltage of 400 V. The stimulation protocol included a high frequency doublet (double pulses with inter-pulse intervals: 10 ms, 100 Hz) that was superimposed during MVC and a potentiated high frequency doublet (Db) and a single stimulus (St) delivered 3 and 6 s, respectively, after the end of the MVC in a resting state.

Then, we calculated the VA as previously proposed [20], which served as an index of central fatigue.

### 2.4. Electromyographic Measurements

Pre-gelled surface electrodes (circular contact area of 1 cm, inter-electrode distance of 2 cm, Ambu Blue Sensor, N-00-S/25, Ballerup, Denmark) were placed over the pre-prepared skin above the VL approximately 5 cm above the lateral side of the patella [21]. Prior to the application of the electrodes, we shaved and cleaned the skin with abrasive paste. Before starting the measurements, an operator checked that there was low tissue resistance between each electrode pair (impedance < 10 kΩ). EMG signals were sampled at a frequency of 2 kHz and recorded by an electromyographic system (MP 100, Biopac Systems Inc., Goleta, CA, USA). Then, we analyzed the data by using LabChart 6 (ADInstruments). VL-EMG responses (peak-to-peak Mmax) to electrical stimulation were further analyzed as a measure of peripheral changes in conjunction with force outputs.

### 2.5. Physiological Measurements during the Downhill Running Trial

Before starting the DR, we asked the participants to perform a 10-min warm up at a self-selected speed. For the DR, the treadmill was inclined at a −25% slope, and the speed was adjusted to the speed corresponding to ~60% of V̇O_2_max. During the warm-up and throughout the entire downhill test, the inspired and expired air was analyzed by a metabolimeter previously calibrated (as described above).

### 2.6. Rate of Perceived Exertion (RPE)

During the downhill running trial, we asked the participants to report fatigue and pain sensation every five minutes. The participants had to express their general fatigue, pain perception in the leg and dyspnea by using the Borg CR-10 Scale [22], with a 0 value meaning ‘nothing at all’ and a 10 value meaning ‘extremely strong’.

### 2.7. Contact and Flight Time

We measured contact and flight time every five minutes during DR using a digital camera with a sample frequency of 400 Hz (Nikon J1, Minato, Japan). We analyzed 10 subsequent steps using Kinovea 0.8.15 software (www.kinovea.org) [23]. In our analysis, we used the mean values of contact and flight time over 10 steps for calculating step frequency (SF, step/s) as SF *= 1/(t_c +_ t_a_)* and step length (SL, m) as SL = running speed/SF.

### 2.8. Statistical Analysis

We analyzed the data using GraphPad Prism 8.0 with alpha set to *p* ≤ 0.05. We performed the ROUT method with a Q = 1% [24] to detect any outliers in all parameters. We performed the Shapiro-Wilk normality test for all the presented parameters. We analyzed the HR, O_2_ drift and RPE at minute 3, 6, 9, 12, and 15 using repeated measures one-way ANOVA. As well, we used one-way ANOVA for analyzing the biomechanical parameters (step frequency and step length). We investigated neuromuscular parameters using two-tailed Student’s *t*-tests between PRE and POST, with the exception of VA, which was analyzed with the Wilcoxon signed-rank test (one-tailed) because the distribution was not Gaussian. As well, we calculated percent changes from PRE to POST for all neuromuscular parameters. Additionally, we calculated the effect sizes (ES) using Cohen’s *d* (0 < *d* < 0.20 *small*; 0.20 < *d* < 0.50, *medium*; 0.50 < *d*, *large*). Finally, we tested the linear correlation coefficient between V̇O_2_, HR drift and changes in neuromuscular parameters was present.

## 3. Results

### 3.1. Downhill Running Trial

The average speed chosen by the participants during the DR trial was 14.6 ± 2.2 km/h, which means a vertical velocity of −3522 ± 522 m/h. The relative intensity of the DR trial was 58.3 ± 6.7% of the V̇O_2_max at minute three and 67.8 ± 12.3% at minute fifteen.

VO_2_ increased as a function of time and at min 15 was +19.8 ± 15.9% higher (*p* < 0.001; ES: 0.49, *medium)* than at min 3 (Figure 2). In addition, HR (in beats per minute) increased as a function of time and was +10.4 ± 10.0% higher at min 12 (*p* < 0.001; ES = 0.44, *medium*) and +15.3 ± 9.9% higher at min 15 (*p* < 0.001; ES: 0.55, *large*) than at min 3 (Figure 2).

### 3.2. Maximal Voluntary Contraction Torque and Voluntary Activation

After the DR, the KE MVCt decreased from 262.9 ± 51.0 Nm to 202.7 ± 42.7 Nm −22.2 ± 12.0%; *p* < 0.001; ES = 0.55, *large*, Figure 3A). The torque evoked by the high frequency doublet after the end of MVC decreased from 86.2 ± 14.2 Nm to 62.8 ± 16.3 Nm (−28.5 ± 12.7%; *p* < 0.001; ES: 0.61, *large*, Figure 3B). The torque evoked by the singlet decreased from 54.3 ± 11.7 Nm to 32.4 ± 10.3 Nm (−41.6 ± 13.6%; *p* < 0.001; ES: 0.70, *large*, Figure 3C). The Mmax evoked by the singlet decreased from 9.9 ± 3.1 mV to 8.6 ± 2.6 mV (−11.8 ± 12.1%; *p* < 0.001; ES: 0.22, *small*, Figure 3D). No differences (*p* = 0.073; ES: 0.14, small) were detected in voluntary activation between PRE and POST.

### 3.3. Perceived Exertion

General, dyspnea and leg RPE increased throughout the DR trial (Figure 4). One-way ANOVA revealed a significant increase (time effect: *p* < 0.001) in all these parameters. Specifically, general RPE increased from 1.8 ± 1.2 at minute three to 5.7 ± 2.4 at minute 15 (*p* < 0.001; ES: 0.71, *large*; Figure 4A). Dyspnea RPE increased from 1.9 ± 1.0 at minute 3 to 4.2 ± 2.1 at minute 15 (*p* < 0.001; ES: 0.57, *large*; Figure 4B). Leg RPE increased from 1.9 ± 1.1 at minute 3 to 6.2 ± 2.3 at minute 15 (*p* < 0.001; ES: 0.76, *large*; Figure 4C). We found no correlations between the RPE increases and other analyzed parameters.

### 3.4. Correlation between Oxygen Drift and MVCt

We found a low negative correlation between the O_2_ drift and the loss in MVCt (*p* = 0.057; r = 0.47; CI = −1.35 to 0.02) (Figure 5). Additionally, we found no correlations with neuromuscular and biomechanical parameters.

### 3.5. Mechanical Parameters

The step frequency was 175 ± 15, 176 ± 14, and 179 ± 18 steps/min after 5, 10, and 15 min, respectively. However, statistical analysis did not reveal differences in time (*p* = 0.158). Consequently, we found no differences in step length (*p* = 0.315), which was 1.45 ± 0.28, 1.45 ± 0.29, and 1.42 ± 0.31 m after 5, 10, and 15 min, respectively.

## 4. Discussion

The main findings of the present study show that 15 min of treadmill DR at a slope of −25% at ~60% of V̇O_2_max 1) lead to a decrease in the MVCt, mainly due to peripheral alterations; 2) induce V̇O_2_ and HR drifts that are not strictly related to MVCt loss.

One of the new aspects of this research is that we combined metabolic measurements with neuromuscular assessment during and after a steep DR exercise. Previous studies evaluated these aspects separately and it was necessary to merge these measurements. Only Vernillo et al. [25] analyzed the metabolic and neuromuscular characteristics (plantar flexor muscle MVCt decreased by ~18%) of DR at the same time, but their results were constrained to a specific condition. Similarly, Giandolini et al. [4] reported that ~34 min of outdoor DR (1264 m of elevation drop with an average vertical speed of −0.62 m/s) performed at a slope of −20% leads to a decrease in the KE-MVCt of ~19%. Analogous knee extensor force loss was assessed after an intermittent 30 min DR completed on an average slope of −12% [26]. We reported a similar value of MVCt loss, but the duration of the trial was less than half of that reported in these studies [4,26] with a much higher vertical speed (−0.95 m/s). The difference in vertical speed is likely be attributed to the different protocols used (outdoor vs. treadmill) and to the different fitness levels of the participants involved in these studies. We supposed that athletes who completed DR at a higher speed adopted a longer step length that can be associated with the loss in muscle force. Indeed, authors [27] reported that there is a length-dependent muscle component in the development of fatigue after eccentric exercise. Thus, athletes who run with longer step lengths may experience higher exercise-induced muscle damage (EIMD) and MVCt loss. However, the faster athletes did not show higher loss in MVCt (*p* > 0.05). Indeed, counter to our hypothesis, we did not find a correlation between running speed and MVCt loss (*p* = 0.972). It is possible that athletes who run at faster speeds might have obtained greater adaptations during their own training sessions and, hence, they experienced lower EIMD during the present test. It would be interesting to further investigate whether athletes might show different MVCt losses when different step length strategies are chosen at the same DR speed. In theory, athletes with higher step frequencies and shorter step lengths should suffer less fatigue and lower MVCt loss after the same DR effort [28,29]. The subsequent practical application would be that increasing step frequency rather than step length would become useful for athletes to minimize muscle damage during DR sections. However, in contrast to other authors [29], we found no correlation between MVCt loss and biomechanical parameters.

In our study, the acute loss in MVCt after DR is mainly explained by peripheral alterations, as shown by the greater impairments in Db, St, and Mmax. The VL Mmax amplitude decreased by ~12% in the present study. Similar results have been observed either after the end of a 30-km trail run (−10% slope) [30] or after the completion of a 6.5 km long downhill section (−16% slope) [4]. The decrease in the Mmax can be ascribed to a reduction in sarcolemmal excitability, which affects muscle fiber activation by altering the generation and propagation of the action potential along the sarcolemma [31]. Parallel to changes in Mmax amplitude, we found lower Db and St after DR, consistent with previous research [4]. Db and St reduction after exercise might suggest the presence of alterations in the contracting apparatus within the muscle [32]. Furthermore, repeated eccentric contractions might cause an acute impairment in the E-C coupling system, also called ‘low frequency fatigue’ (LFF). Previous studies have shown that this mechanism involves impairments at some levels within the cascade of E-C events: decreased calcium release from the sarcoplasmic reticulum and/or myofibrillar calcium sensitivity [32]. LFF mechanism could have had an impact in determining the MVC loss after DR as already reported by Giandolini et al. [4]. Nevertheless, exercise-induced MVC force loss might also have a central origin (i.e., reduction in voluntary activation) [13]. Impairments might arise at the spinal and/or supraspinal level. We did not investigate the precise localization of these alterations since it would have required a more complex set up. We focused on analyzing VA by using the twitch interpolation technique before and after DR and found a non-significant reduction by only ~2%. A similar downhill exercise protocol produced greater and significant VA changes (−7%) [4]. Similarly, when longer trail running performances are considered, authors have reported greater VA impairments (−19%, −24%) [14,20]. Different fatiguing exercises (i.e., exercise volume, intensity, and type of contractions involved in the task) and neuromuscular evaluation collection time (i.e., the time delay at which the neuromuscular performance was evaluated after the end of the exercise) might contribute to generate different degrees of alteration in VA. For this reason, we acknowledge that the 10 min delay between the end of the DR and the evaluation of the neuromuscular function in the protocol of the present study might have underestimated the central alteration in fatigue-induced force loss. The neuromuscular evaluation responses shown by the participants in our study might suggest that in the acute phase after the cessation of the DR protocol, they might have been capable of recruiting almost all the motor units necessary to maximize their force. However, predominantly due to several peripheral impairments, the maximization of their force output was not possible. Indeed, EIMD and fatigue developed during repeated eccentric contractions might have affected the force expression [26]; this is probably true even in well-trained mountain running athletes [1,3,20]. The mechanical stress that the muscles experienced during this type of exercise might have generated a disruption in the sarcomeres and, eventually, degeneration of muscle fibers. However, other factors should be considered, such as metabolic depletion, calcium influx, generation of reactive oxygen species, and musculotendinous stiffness regulation (see the review by Byrne et al. [33] for details).

Loss in MVCt was not strictly related to V̇O_2_ drift (*p* = 0.057), even if there was a low negative association. Other authors [34] were also unable to find a relationship between muscle damage and V̇O_2_ drift, suggesting that the loss in MVCt is not a relevant factor in determining the V̇O_2_ drift. The different results obtained comparing other studies [6,8] may be ascribed to the different protocols and different levels of the participants implemented. However, future studies with a greater number of participants may confirm this relationship.

We acknowledge that our study has some limitations. First, it was conducted in the laboratory, and the results may be different compared to outdoor conditions. In particular, running on a treadmill or overground have different biomechanical requirements. Also, the duration of the trial was shorter compared to ‘classic’ trail running races. In addition, it would be interesting to assess athletes with different skills in downhill running to understand what determines the performance in this population. Also, it would be interesting to understand the real importance of downhill performance in the overall race. In fact, it is common for athletes who are not the best in uphill to be able to gain several positions during downhill sections. However, a different study design would be required for this type of evaluation.

## 5. Conclusions

In conclusion, we found that after 15 min of DR, athletes experience a decrease in MVCt, which is ascribed mainly to peripheral rather than central alterations. Additionally, during low-intensity DR exercise, V̇O_2_ and HR drifts occur that are not associated with the loss in MVCt. Although it was not the aim of this study to analyze the mechanics of downhill running, based on the results of the present and previous studies [28,29] we suggest that athletes should: (1) adopt higher step frequency (instead of step length) when they want to increase the running speed during downhill sections; and (2) maintain a low HR to limit the increase in V̇O_2_ slow component and the fatigue effects. In addition, proper recovery and nutrition strategies could also influence final performance.

## Figures and Tables

**Figure 1 ijerph-18-08135-f001:**
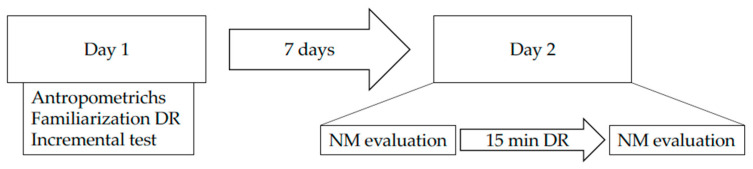
Experimental design. NM: neuromuscular; DR: downhill running.

**Figure 2 ijerph-18-08135-f002:**
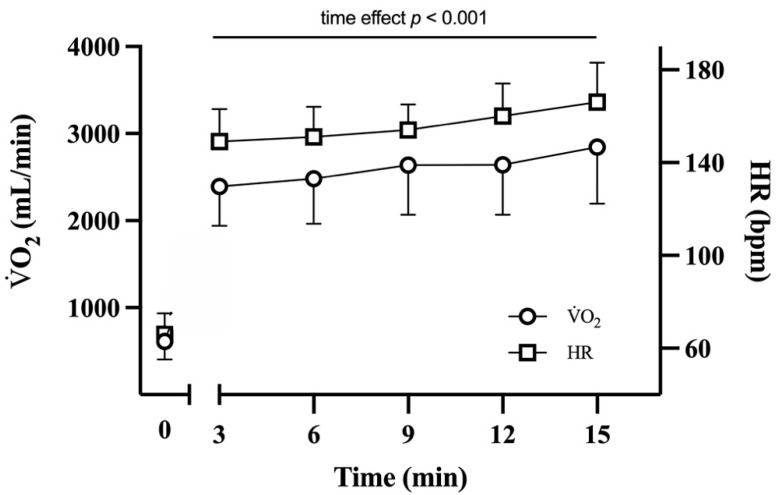
V̇O_2_ and HR vs. time. Mean (and SD) V̇O_2_ and HR during the 15-min downhill running. V̇O_2_: oxygen consumption; HR: heart rate. *p* represents the significant ‘time effect’ obtained by repeated measures one-way ANOVA (see “Results”).

**Figure 3 ijerph-18-08135-f003:**
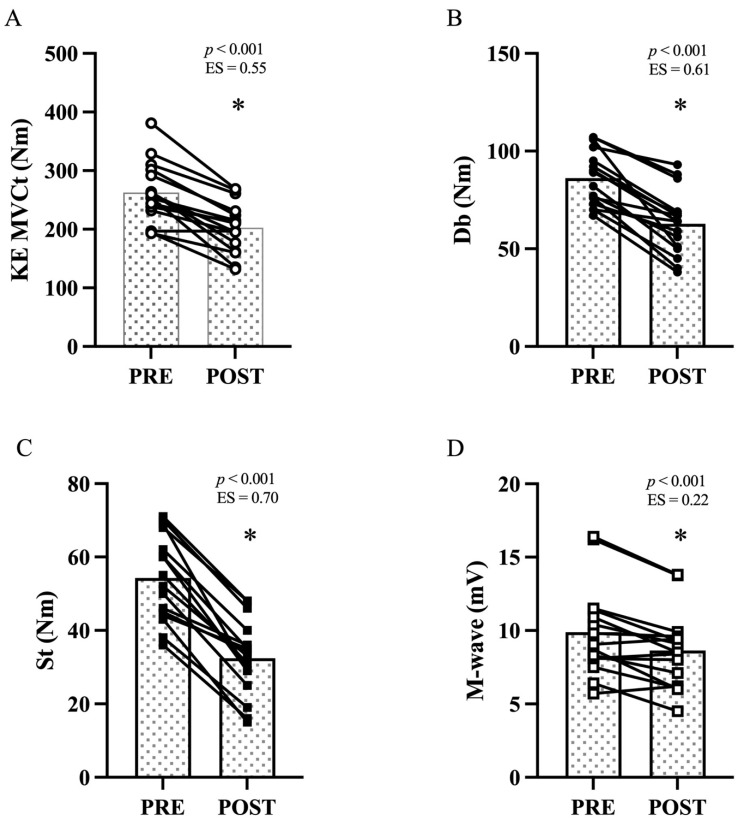
Maximal voluntary contraction torque and muscle contractility. Mean (dotted columns) and individual values (black dots) for VL before (PRE) and after (POST) the 15-min downhill run (n = 20). (**A**) KE maximal voluntary torque; (**B**) Maximal evoked torque by high frequency doublet (Db, 100Hz) on the KE muscles; (**C**) KE torque evoked by the single pulse stimulation (St); (**D**) Mmax peak-to-peak amplitude. * *p* < 0.001 compared with PRE.

**Figure 4 ijerph-18-08135-f004:**
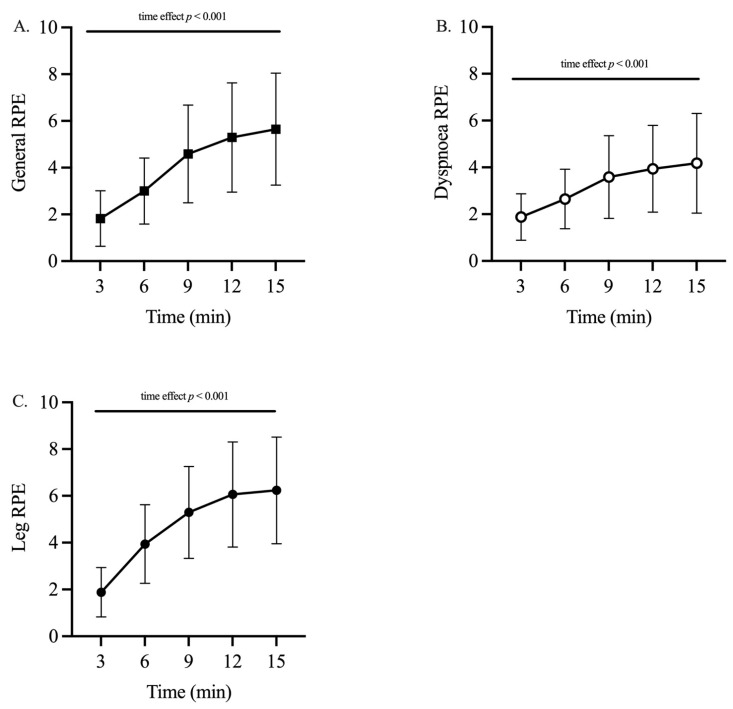
Rate of perceived exertion. General (**A**), dyspnoea (**B**), and leg (**C**) RPE of the participants (n = 22) collected every five minutes. *p* represents the significance obtained by repeated measures one-way ANOVA (see “Results”).

**Figure 5 ijerph-18-08135-f005:**
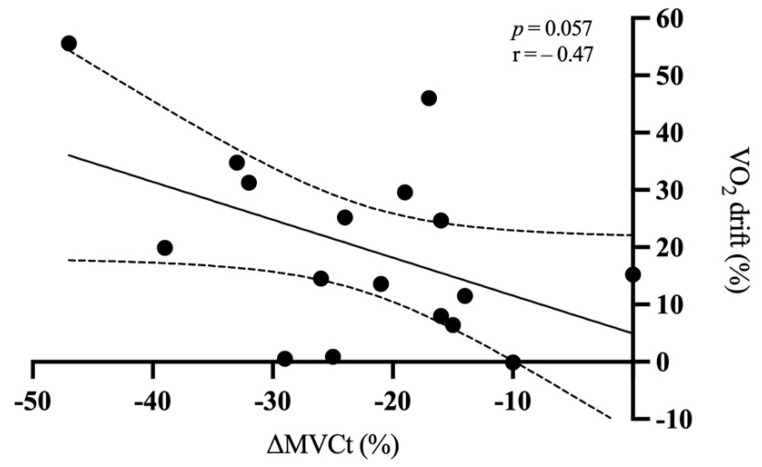
Relationship between MVCt decrease and O_2_ drift (in %). Dotted lines represent 95% confidence interval (n = 17). V̇O_2_: oxygen consumption; MVCt: maximal voluntary contraction torque.

## Data Availability

All the data are stored by the corresponding author and can be required by contacting him at nicola.giovanelli@uniud.it.

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
