# Peer review of "Peripheral Alterations Affect the Loss in Force after a Treadmill Downhill Run"

_ijerph, 2021, doi:10.3390/ijerph18158135_

Round 1

Reviewer 1 Report

Introduction:

The introduction is correctly written, the bibliography presented is adequate and a common thread is followed.

Material and methods:

I suggest adding a figure that clarifies the evaluation procedure. Even so, the material and methods section is well developed. 

Line 65: Present project registration number in the ethics committee. 

Results:

The results are clearly presented. 

Discussion.

What were the limitations of the study?
What are the practical applications of the study?

Reviewer 2 Report

The authors have investigated the cardiorespiratory response during 15 minutes of downhill running to evaluate the neuromuscular consequences in a group of trail runners. They have found that after 15 min of DR, athletes had a decreased MVCt, which was ascribed mainly to peripheral rather than central alterations. They also have proved that during low-intensity DR exercise, muscle fatigue and exercise-induced muscle damage can contribute to the development of O2 and HR drift. Their idea seems new and from a technical point of view has implemented very good which is worth publishing. However, the paper needs proofreading as there are several grammatical problems in the paper.  Abstract is well written, topics have meaning full insight and useful for the future research as well as relevant. The writing of the paper is very good. After all this. The paper is acceptable for the publication.

Reviewer 3 Report

Line 215 probably 3.5

The conclusion should be completed by highlighting the practical consequences for training and recovery after a downhill trail.

I was very interested in this research work on the downhill trail. The general opinion is that the trail is won in the descent and many trainers insist on the shorter and more frequent stride which will allow to limit the force of the impacts at each support and to control its supports for more stability. The authors must simply complete the conclusion by insisting on what they can deduce from the research work for the training and coaching of these athletes.  1°) What is the consequence of peripheral muscular alteration, how to improve recovery? 2°) What is the practical consequence of the cardiovascular modifications noted?   The conclusion is often what is read first and deserves to be better adapted to sport professionals.  
